# Hallu-PI: Evaluating Hallucination in Multi-modal Large Language Models within Perturbed Inputs

Peng Ding*
National Key Laboratory for Novel
Software Technology,
Nanjing University,
Nanjing, China
dingpeng@smail.nju.edu.cn

Jingyu Wu*
College of Computer Science and
Technology,
Zhejiang University,
Hangzhou, China
wujingyu@zju.edu.cn

Jun Kuang
Meituan
Shanghai, China
kuangjun@meituan.com

Dan Ma
Meituan
Shanghai, China
madan07@meituan.com

Xuezhi Cao
Meituan
Shanghai, China
caoxuezhi@meituan.com

Xunliang Cai
Meituan
Beijing, China
caixunliang@meituan.com

Shi Chen
Zhejiang-Singapore Innovation and
AI Joint Research Lab,
Zhejiang University,
Hangzhou, China
shelleych@zju.edu.cn

Jiajun Chen
National Key Laboratory for Novel
Software Technology,
Nanjing University,
Nanjing, China
chenjj@nju.edu.cn

Shujian Huang†
National Key Laboratory for Novel
Software Technology,
Nanjing University,
Nanjing, China
huangsj@nju.edu.cn

## Abstract

Multi-modal Large Language Models (MLLMs) have demonstrated remarkable performance on various visual-language understanding and generation tasks. However, MLLMs occasionally generate content inconsistent with the given images, which is known as "hallucination". Prior works primarily center on evaluating hallucination using standard, unperturbed benchmarks, which overlook the prevalent occurrence of perturbed inputs in real-world scenarios—such as image cropping or blurring—that are critical for a comprehensive assessment of MLLMs' hallucination. In this paper, to bridge this gap, we propose **Hallu-PI**, the first benchmark designed to evaluate **Hallu**cination in MLLMs within **P**erturbed **I**nputs. Specifically, Hallu-PI consists of seven perturbed scenarios, containing 1,260 perturbed images from 11 object types. Each image is accompanied by detailed annotations, which include fine-grained hallucination types, such as existence, attribute, and relation. We equip these annotations with a rich set of questions, making Hallu-PI suitable for both discriminative and generative tasks. Extensive experiments on 12 mainstream MLLMs, such as GPT-4V and Gemini-Pro Vision, demonstrate that these models exhibit significant hallucinations on Hallu-PI, which is not observed in unperturbed scenarios. Furthermore, our research reveals a severe bias in MLLMs' ability to handle different types of hallucinations. We also design two baselines specifically for perturbed scenarios, namely Perturbed-Reminder and Perturbed-ICL. We hope that our study will bring researchers' attention to the limitations of MLLMs when dealing with perturbed inputs, and spur further investigations to address this issue. Our code and datasets are publicly available at https://github.com/NJUNLP/Hallu-PI.

## CCS Concepts

• **Information systems** → **Multimedia databases**; **Multimedia content creation**.

## Keywords

Multi-modal Large Language Models, Hallucination, Perturbed Inputs, Benchmark Evaluation

**ACM Reference Format:**
Peng Ding, Jingyu Wu, Jun Kuang, Dan Ma, Xuezhi Cao, Xunliang Cai, Shi Chen, Jiajun Chen, and Shujian Huang. 2024. Hallu-PI: Evaluating Hallucination in Multi-modal Large Language Models within Perturbed Inputs. In *Proceedings of the 32nd ACM International Conference on Multimedia (MM '24), October 28-November 1, 2024, Melbourne, VIC, Australia.* ACM, New York, NY, USA, 9 pages. https://doi.org/10.1145/3664647.3681251

*Equal contribution.

†Corresponding author.

## 1 Introduction

Multi-modal Large Language Models (MLLMs) have achieved significant progress in a range of practical applications, such as providing detailed descriptions for user-provided images (i.e., image captioning) [1, 29] and answering specific questions about input images (i.e., visual question answering) [24, 41]. However, these models occasionally exhibit a phenomenon known as "hallucination", where the generated content is inconsistent with the given images [16, 35].

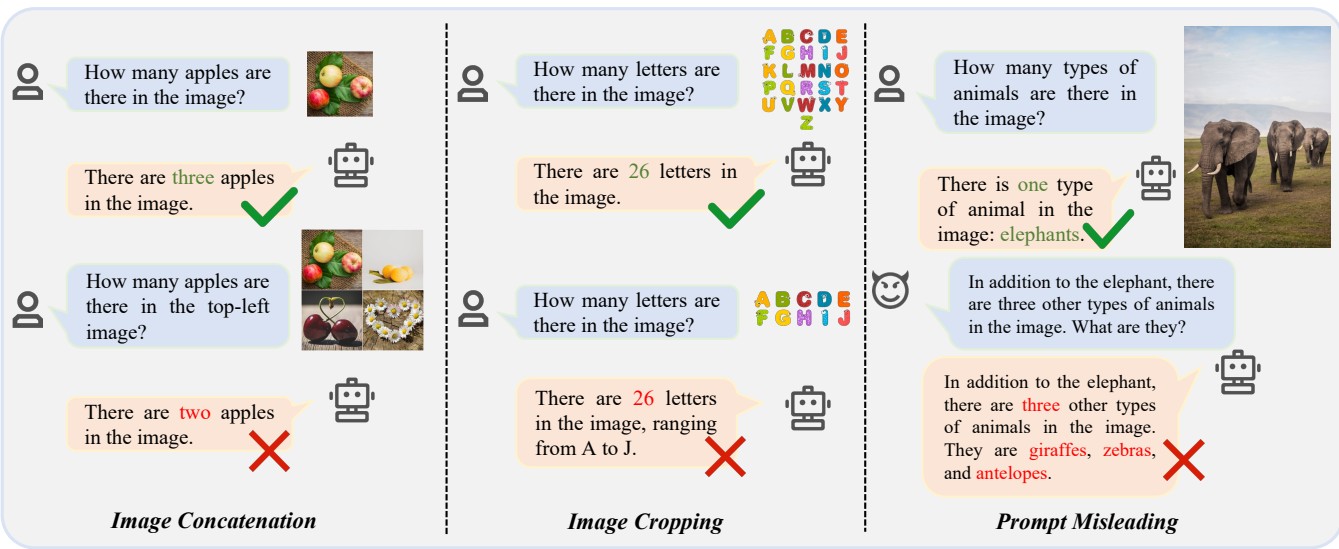

**Figure 1: Some examples of hallucinations in MLLMs with perturbed inputs (such as image concatenation, image cropping, and prompt misleading). Text highlighted in green and red represents correct and hallucinatory content, respectively.**

Previous works have sought to investigate the hallucinations in MLLMs by utilizing a large language model like GPT-4 [1], or by employing humans as annotators [38, 40]. Alternatively, some studies focus on developing detection models to scrutinize the hallucinations exhibited by MLLMs. [14, 22]. More recently, [31] introduce AMBER, a LLM-free benchmark designed to examine MLLM hallucinations in both discriminative and generative tasks across dimensions like existence, attribute, and relation.

Despite these efforts, existing researches primarily focus on conducting evaluations by sampling images from available image datasets, such as MSCOCO [14, 22, 28, 32, 37, 38]. However, in real-world scenarios, inputs fed to MLLMs frequently encounter a variety of perturbations (e.g., noise and cropping) [12]. Overlooking such perturbations could lead MLLMs to produce incorrect answers or judgments, potentially causing serious accidents in certain applications (e.g., medical diagnosis, industrial automation and autonomous driving) [17]. Figure. 2 illustrates the hallucinations of several MLLMs before and after image concatenation perturbation. The inconsistent performance trends indicate that relying solely on existing unperturbed benchmarks is insufficient for a comprehensive and precise evaluation of hallucinations in MLLMs.

In order to bridge this gap, we introduce **Hallu-PI**, a benchmark designed to evaluate the **Hallu**cination performance of MLLMs within **P**erturbed **I**nputs. Followed by [12, 15], we first categorize the image perturbations into four types: noise, blur, weather, and digital. Additionally, we meticulously propose three distinct types of perturbations: image concatenation, image cropping, and prompt misleading. These perturbations are considered at both the image level and the prompt level. Annotators are instructed to carefully manipulate the perturbations and provide corresponding annotations. Evaluations of 12 mainstream MLLMs conducted on Hallu-PI reveal significant hallucinations of leading MLLMs (e.g., GPT-4V and Gemini-Pro Vision) when dealing with perturbed scenarios.

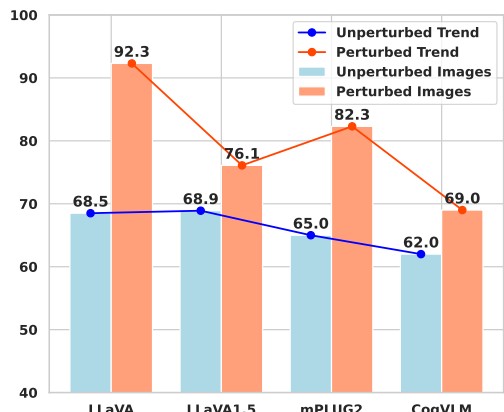

**Figure 2: The hallucinatory performance of various MLLMs before (blue bars) and after (orange bars) input perturbation. Inconsistent performance trends show that relying solely on unperturbed benchmarks is insufficient for a complete and precise evaluation of hallucinations in MLLMs.**

To comprehensively understand the hallucination of MLLMs to perturbed inputs, we conduct a detailed analysis of the experimental results. We find that most models exhibit significant bias towards specific types of perturbations, particularly image concatenation, image cropping, and prompt misleading (see Figure. 1). Furthermore, to mitigate the hallucination of MLLMs in response to perturbed inputs, we draw inspiration from the defensive strategies adopted by text LLMs against jailbreak attacks [7, 34] and designed two baselines: Perturbed-Reminder and Perturbed-ICL. Experiments conducted on GPT-4V show that these strategies effectively reduce hallucinations. We hope our work can prompts MLLM researchers and developers to address hallucinations from perturbed inputs.

**Table 1: Comparison with existing hallucination evaluation benchmarks. "Sample" means sampling from an existing dataset.**

| Benchmark | Task Type | | Hallucination Type | | | Perturbation | Baseline | Source |
|---|---|---|---|---|---|---|---|---|
| | Discriminative | Generative | Existence | Attribute | Relation | | | |
| POPE [22] | ✓ | ✗ | ✓ | ✗ | ✗ | ✗ | ✗ | Sample |
| M-HalDetect [14] | ✗ | ✓ | ✗ | ✗ | ✗ | ✗ | ✓ | Sample |
| HaELM [32] | ✗ | ✓ | ✗ | ✗ | ✗ | ✗ | ✗ | Sample |
| Halle-Switch [38] | ✗ | ✓ | ✗ | ✗ | ✗ | ✗ | ✗ | Sample |
| AMBER [31] | ✓ | ✓ | ✓ | ✓ | ✓ | ✗ | ✗ | Manual |
| Hallu-PI (ours) | ✓ | ✓ | ✓ | ✓ | ✓ | ✓ | ✓ | Manual |

In summary, the contributions of our work are as follows:

- We construct Hallu-PI, the first freely available multi-modal hallucination benchmark with perturbed inputs. Hallu-PI encompasses 7 perturbed scenarios, a total of 1,260 images, and 11 distinct object categories to evaluate hallucinations in MLLMs across both generative and discriminative tasks.
- We conduct extensive experiments with Hallu-PI to evaluate multi-modal hallucinations in 12 state-of-the-art MLLMs under perturbed inputs. The results unveil the limitations of MLLMs when dealing with perturbed inputs, as well as their specific bias towards certain types of hallucinations.
- To mitigate the hallucinations of MLLMs on Hallu-PI, we introduce two baselines: Perturbed-Reminder and Perturbed-ICL. Experimental results on GPT-4V indicate that our methods are effective and can reduce the model's hallucinations in response to perturbed inputs to a certain extent.

## 2 Related Work

### 2.1 Multimodal Large Language Models

Multi-modal Large Language Models (MLLMs) are currently achieving significant improvements by combining the advanced capabilities of Large Language Models (LLMs) with visual processing [1, 2, 9, 21, 29]. These MLLMs show great potential in a variety of applications, such as visual question answering (VQA) [11], image captioning [26], and video understanding [19]. Representative MLLMs, such as CogVLM [33], LLaVA1.5 [24], InternLM-XComposer [39], MiniGPT-4 [5], mPLUG-Owl2 [36], Qwen-VL [3], and the latest GPT-4V [1], and Google Gemini-Pro Vision [29], have achieved impressive performance across various multi-modal tasks.

### 2.2 Hallucination in MLLMs

While MLLMs have exhibited excellent performance on multi-modal tasks, we are still facing the challenge that MLLMs often generate content unfaithful to the given images, which is called "hallucination" [6, 18, 20, 23, 30].

Currently, many researchers focus on evaluating the hallucination in MLLMs. LURE [40] and HallE-Switch [38] rely on human evaluations or GPT-4. While this method is relatively reliable, it is also expensive. HaELM [32] and FDPO [14] are based on hallucinatory detection models. However, the performance of these models is highly dependent on hallucinatory data and incurs substantial training costs. POPE [22] is based on object detection but is only applicable to discriminative tasks and evaluates existence-type hallucinations. Recently, [31] introduce AMBER, which assesses

hallucinations across multiple dimensions, such as existence, attribute, and relation. Despite these efforts, they do not explore hallucinations in the perturbed scenarios commonly encountered in real-life situations. To bridge this gap, we propose Hallu-PI, the first benchmark designed to evaluate the hallucination of MLLMs with perturbed inputs. Table. 1 presents a detailed comparison between Hallu-PI and other hallucination benchmarks.

### 2.3 Image Perturbation

To simulate real-world perturbation scenarios, previous works adopt various perturbation strategies such as ImageNet-C [15] and Stylize-ImageNet [12, 27, 28]. The perturbations are grouped into five primary categories: noise, blur, weather, digital, and stylize. Specifically, these can be further subdivided into the following 17 image perturbation techniques: (1) Noise: Adding noise to the images, such as gaussian noise, shot noise, impulse noise, and speckle noise. (2) Blur: Blurring the images, including defocus blur, frosted glass blur, motion blur, and zoom blur. (3) Weather: Adding environmental effects such as snow, frost, fog, and brightness adjustments. (4) Digital: Manipulating images through contrast enhancement, elastic transformation, pixelation, and JPEG compression. and (5) Stylize: Applying artistic styles and transformations to images.

Compared to existing benchmarks that only consider hallucination assessment in unperturbed scenarios, Hallu-PI further takes into account perturbations that frequently occur in real-world applications. Therefore, it serves as a complement to existing benchmarks and provides a more comprehensive and accurate evaluation of hallucinations in MLLMs.

## 3 Hallu-PI Benchmark

In this section, we introduce the process of constructing our Hallu-PI benchmark which primarily encompasses three aspects: (1) Image Collection, (2) Image Perturbation and Annotation, and (3) Designing Prompt Query Templates.

### 3.1 Image Collection

To ensure the diversity of the dataset, we identify 11 different object types and require annotators to collect images for each category. In the image selection process, we primarily consider (1) image copyright and (2) image quality. We provide annotators with several websites offering free copyright images and instruct them to search for images using specific object keywords. Annotators are asked to select images where the object is complete and the image is of high quality for downloading.

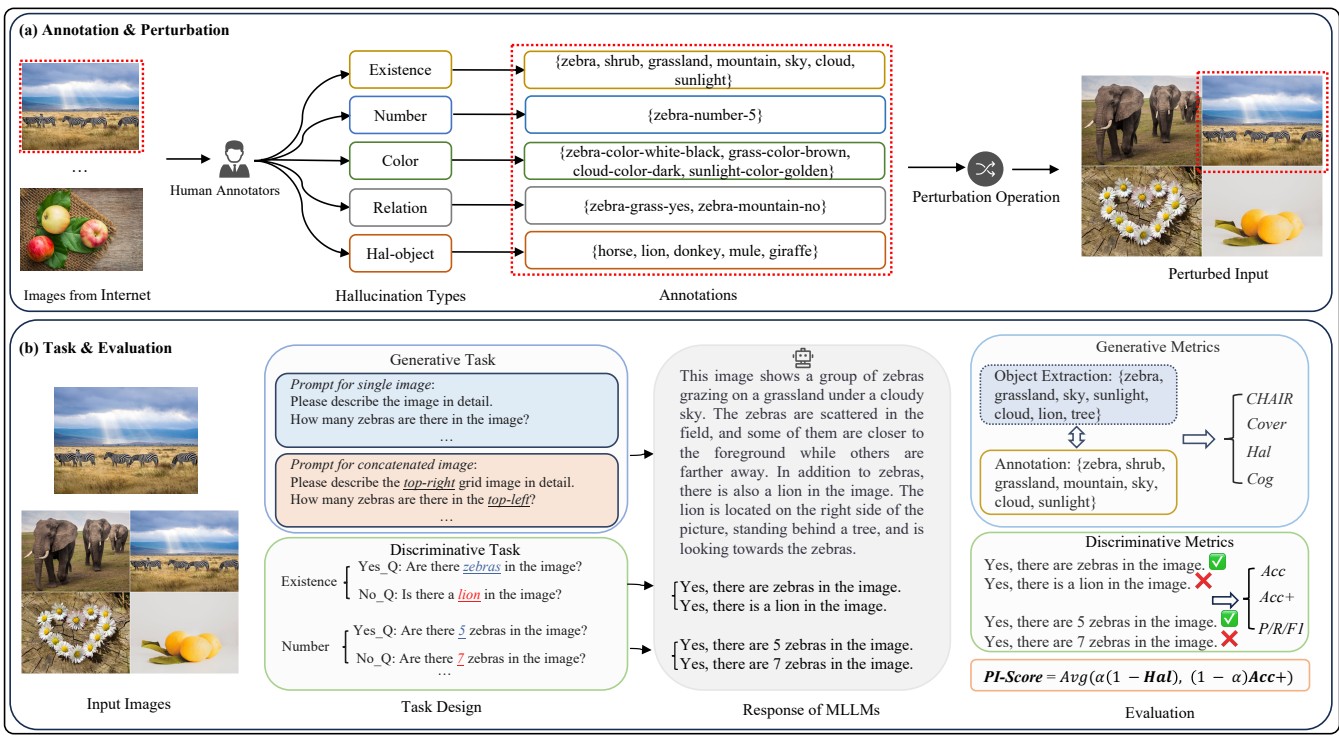

**Figure 3: (a) Overview of Hallu-PI pipeline for image annotation and perturbation. (b) An illustration of evaluation pipeline of Hallu-PI, including both generative and discriminative tasks.**

## 3.2 Image Perturbation and Annotation

Following previous work [28], we first consider four primary types of perturbation: noise, blur, weather, and digital. Stylize is not included because the stylized images are too blurred to recognize the objects and attributes within them. To construct a more comprehensive set of perturbation scenarios, we meticulously propose three additional perturbations: image concatenation, image cropping, and prompt misleading. The first two are considered because they are commonly used by users in real life to edit their images, while prompt misleading ensures that Hallu-PI can evaluate hallucinations at both the image level and the prompt level.

For noise, blur, weather, and digital perturbations, we reuse the code from [28] to generate the perturbed images. For image concatenation, we require our well-trained annotators to combine every four individual images previously collected into a single four-grid image, ensuring that the objects in the concatenated image are complete. For image cropping, we primarily focus on images containing English letters. Annotators are instructed to crop these images and provide corresponding questions and answers for both the original and cropped images. For prompt misleading, annotators need to select an image and provide a prompt that could potentially induce hallucinations. Figure. 1 provide some examples of these perturbations. Annotators are required to provide detailed annotations for perturbed images. These annotations include Existence, Number, Color, Relation, and Hal-object, as shown in Figure. 3.

In Figure. 4, we present the distribution of perturbation types and the distribution of object categories included in Hallu-PI.

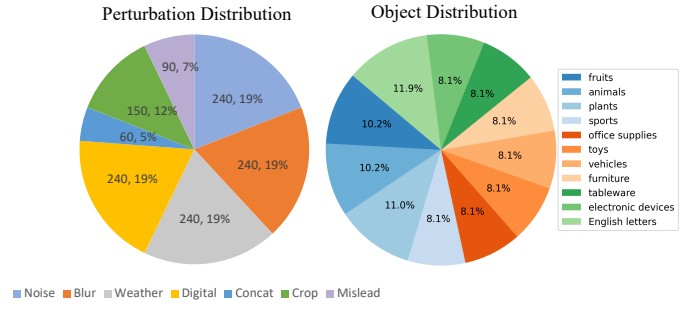

**Figure 4: The data distribution of Hallu-PI.**

## 3.3 Designing Prompt Query Templates

To ensure a more comprehensive evaluation of hallucinations, we design both generative and discriminative prompt templates for each perturbation scenario. For the perturbations such as noise, blur, weather, digital, and image concatenation, we pose questions regarding each specific annotation field. For instance, for the image concatenation perturbation, the generative prompt for the "Existence" field before perturbation is: "Please describe the *existing objects* in the image." After perturbation, the prompt becomes: "Please describe the *existing objects* in the *top-left* image." with "*existing objects*" and "*top-left*" being flexible and variable. For the design of discriminative prompts, we consider that merely calculating accuracy might be insufficient. Following previous work [11], we design

**Table 2: The results under noise, blur, weather, and digital perturbations. Before/After means before/after perturbation.**

| Model | Before | | After | | | | | | | |
| | | | Noise | | Blur | | Weather | | Digital | |
| | ACC+↑ | CHAIR↓ | ACC+↑ | CHAIR↓ | ACC+↑ | CHAIR↓ | ACC+↑ | CHAIR↓ | ACC+↑ | CHAIR↓ |
|---|---|---|---|---|---|---|---|---|---|---|
| CogVLM | 49.0 | 62.0 | 48.5 | 68.2 | 47.4 | 68.6 | 42.8 | 67.9 | 48.4 | 69.8 |
| Multi-GPT | 13.3 | 73.5 | 9.6 | 73.6 | 12.8 | 76.1 | 11.2 | 73.4 | 9.2 | 77.8 |
| LLaVA | 6.3 | 68.5 | 4.33 | 67.7 | 5.0 | 70.6 | 4.17 | 69.8 | 3.6 | 74.2 |
| LLaVA1.5 | 43.0 | 68.9 | 42.6 | 70.1 | 42.4 | 68.7 | 43.3 | 68.0 | 36.8 | 74.5 |
| MiniGPT-4 | 16.0 | 72.4 | 15.8 | 70.2 | 15.9 | 72.1 | 14.5 | 72.6 | 13.8 | 73.9 |
| MiniGPT4-v2 | 28.3 | 72.1 | 26.7 | 74.7 | 28.8 | 74.0 | 28.2 | 72.8 | 27.1 | 74.9 |
| mPLUG2 | 38.0 | 65.0 | 33.3 | 67.6 | 33.1 | 69.1 | 35.3 | 66.9 | 32.3 | 73.6 |
| Gemini | 46.0 | 57.3 | 44.2 | 60.0 | 45.1 | 59.7 | 44.8 | 58.5 | 37.5 | 61.3 |
| GPT-4V | 47.3 | 66.1 | 42.3 | 66.9 | 41.8 | 68.4 | 47.8 | 60.9 | 34.0 | 65.4 |

**Table 3: The results under image concatenation, image cropping, and prompt misleading perturbations.**

| MLLMs | PI-Score↑ | | | | | |
| | Concat | | Cropping | | Prompt Mislead | |
| | Before | After | Before | After | Before | After |
|---|---|---|---|---|---|---|
| CogVLM | 45.4 | 22.5 | 10.0 | 5.0 | 39.6 | 11.4 |
| Multi-GPT | 8.3 | 15.0 | 11.7 | 0.0 | 18.9 | 7.2 |
| LLaVA | 6.5 | 2.2 | 3.4 | 6.7 | 14.4 | 5.2 |
| LLaVA1.5 | 32.4 | 5.9 | 10.0 | 8.4 | 26.4 | 8.1 |
| MiniGPT-4 | 8.9 | 5.9 | 10.0 | 8.4 | 18.5 | 7.0 |
| MiniGPT-v2 | 15.8 | 12.3 | 16.7 | 15.0 | 26.4 | 11.3 |
| mPLUG2 | 25.7 | 18.9 | 10.0 | 8.3 | 29.7 | 15.7 |
| InternLM | 38.3 | 37.3 | 8.3 | 10.0 | 34.4 | 28.0 |
| Qwen-VL | 46.3 | 19.6 | 20.0 | 11.7 | 53.2 | 38.2 |
| VisualGLM | 6.8 | 0.6 | 34.0 | 0.0 | 21.2 | 11.3 |
| Gemini | 44.6 | 21.4 | 45.0 | 26.7 | 59.2 | 39.4 |
| GPT-4V | 42.0 | 18.0 | 43.4 | 30.0 | 61.4 | 48.2 |

Yes_Q and No_Q, representing questions with the single-word answers "Yes" or "No," respectively. This allows for the calculation of Acc+, which further enhances the accuracy of hallucination assessment, as shown in Figure. 3.

For the image cropping and prompt misleading perturbations, the generative and discriminative prompts are meticulously designed by the annotators and reviewed by two experts. We present the detailed prompt templates in the supplementary materials.

## 4 Experiments

In this section, we conduct extensive experiments to evaluate the performance of different state-of-the-art MLLMs on our Hallu-PI benchmark. We introduce the primary setup of our experiments, including baseline models (Sec. 4.1), response processing (Sec. 4.2), and evaluation metrics (Sec. 4.3).

### 4.1 Baseline Models

We select multiple mainstream state-of-the-art MLLMs for evaluation, including GPT-4V [1], Google Gemini-Pro Vision [29], InternLM-XComposer-VL [39], QWen-VL-Chat [3], VisualGLM [10], mPLUG-Owl-2 [36], MininGPT4-v2 [5], MiniGPT-4 [41], LLaVA1.5 [24],

LLaVA [25], CogVLM [33], and MultimodalGPT [13]). All models have been fine-tuned on their instruction tuning datasets. To ensure optimal performance, we use the hyper-parameters provided in the official code repositories of the models to generate responses. More details about these MLLMs are in supplementary materials.

### 4.2 Response Processing

The input for Hallu-PI is defined as: $Input = \{Img, Ins\}$, where $Img$ represents the image, and $Ins$ refers to the prompt. As shown in Figure 3, we obtain an initial response $Res$ by inputting $Input$ into a specific MLLM and extracting key elements for computing metrics.

For the generative task, we use the natural language toolkit (NLTK) [4] as an answer extractor to obtain the initial prediction's result $R'_{obj} = \{R_1, R_2, ..., R_n\}$. Then, we construct an objects list $X_{obj} = \{X_1, X_2, ..., X_n\}$ consisting of all annotated objects in Hallu-PI. $X_{obj}$ is used to filter out unnecessary objects in $R'_{obj}$ such as "picture," "distance," and "side." Finally, we obtain the final objects $R_{obj}$ by using $R_{obj} = R'_{obj} \cap X_{obj}$.

For the discriminative task, owing to our prompt template design, "Please answer with 'Yes' or 'No'," we can easily perform quantitative statistics based on the "Yes" or "No" responses included in the MLLM outputs, which is both accurate and objective.

### 4.3 Evaluation Metrics

We first introduce the metrics used for generative task and discriminative task. Then, we present our proposed PI-Score metric, which is a comprehensive metric for evaluating both tasks.

#### 4.3.1 Metrics on Generative Task.

**CHAIR.** CHAIR evaluates the frequency of hallucinatory objects appearing in the responses, which is the most commonly used metric for evaluating hallucinations in MLLMs on generative tasks. With a provided annotated ground truth list $A_{obj} = \{A_1, A_2, ..., A_n\}$, the calculation formula is as follows:

$$\textbf{CHAIR}(Res) = 1 - \frac{len(R_{obj} \cap A_{obj})}{len(R_{obj})} \quad (1)$$

**Cover.** Cover quantifies the degree of correspondence between responses and the image description. Precisely, its value indicates the coverage of objects mentioned in response $R_{obj}$ relative to

**Table 4: The results of generative task on image concatenation, cropping, and prompt misleading.**

| MLLMs | Image Concatenation | | | | | | | | Image Cropping | | Prompt Misleading | |
| | CHAIR ↓ | | Cover ↑ | | Hal ↓ | | Cog ↓ | | Hal ↓ | | Hal ↓ | |
| | Before | After | Before | After | Before | After | Before | After | Before | After | Before | After |
|---|---|---|---|---|---|---|---|---|---|---|---|---|
| CogVLM | 62.0 | 69.0 | 55.3 | 48.3 | 58.3 | 97.1 | 4.3 | 5.9 | 80.0 | 90.0 | 36.7 | 93.3 |
| Multi-GPT | 73.5 | 97.5 | 22.5 | 2.0 | 96.7 | 86.3 | 30.8 | 77.1 | 76.7 | 100.0 | 63.3 | 93.3 |
| LLaVA | 68.5 | 92.3 | 38.8 | 7.4 | 93.3 | 96.7 | 4.3 | 14.9 | 93.3 | 86.7 | 66.7 | 93.3 |
| LLaVA1.5 | 68.9 | 76.1 | 43.8 | 25.0 | 78.3 | 96.3 | 3.4 | 5.7 | 86.7 | 90.0 | 63.3 | 90.0 |
| MiniGPT-4 | 72.4 | 89.3 | 46.5 | 24.8 | 98.3 | 95.8 | 5.1 | 8.2 | 80.0 | 83.3 | 63.3 | 93.3 |
| MiniGPT-v2 | 72.1 | 88.9 | 49.6 | 32.5 | 100.0 | 96.7 | 4.0 | 7.1 | 93.3 | 93.3 | 53.3 | 93.3 |
| mPLUG2 | 65.0 | 82.3 | 44.6 | 14.3 | 86.7 | 89.6 | 6.2 | 6.4 | 93.3 | 96.7 | 46.7 | 80.0 |
| InternLM | 58.4 | 79.2 | 16.3 | 9.5 | 71.7 | 62.5 | 18.8 | 16.7 | 86.7 | 86.7 | 43.3 | 63.3 |
| Qwen-VL | 58.2 | 56.3 | 35.8 | 32.3 | 46.7 | 79.2 | 9.8 | 11.1 | 83.3 | 93.3 | 6.7 | 16.7 |
| VisualGLM | 76.9 | 89.1 | 45.0 | 29.6 | 100.0 | 99.2 | 4.4 | 9.2 | 93.3 | 100.0 | 46.7 | 66.7 |
| Gemini | 57.3 | 63.4 | 50.2 | 43.7 | 56.7 | 90.8 | 3.6 | 4.5 | 26.7 | 56.7 | 12.1 | 30.0 |
| GPT-4V | 66.1 | 63.6 | 66.6 | 53.6 | 63.3 | 98.3 | 1.6 | 1.9 | 33.3 | 73.3 | 1.1 | 3.3 |

manually annotated objects $A_{obj}$:

$$\mathbf{Cover}(Res) = \frac{len(R_{obj} \cap A_{obj})}{len(A_{obj})} \qquad (2)$$

**Hal**. Hal represents the proportion of responses with hallucinations. For a MLLM's response $Res$, if its **CHAIR**$(Res) \neq 0$, then $Res$ is considered to contain hallucinations:

$$\mathbf{Hal}(Res) = \begin{cases} 1, & \text{if} \quad \mathbf{CHAIR}(Res) \neq 0 \\ 0, & \text{if} \quad \mathbf{CHAIR}(Res) = 0 \end{cases} \qquad (3)$$

**Cog**. Cog aims to measure the ratio between hallucinations produced by MLLMs and those annotated by humans. Similar to [31], we use the hallucinatory target list $H_{obj} = \{H_1, H_2, ..., H_n\}$ (corresponding to Hal-object in Figure. 3) to calculate Cog:

$$\mathbf{Cog}(Res) = \frac{len(R_{obj} \cap H_{obj})}{len(R_{obj})} \qquad (4)$$

*4.3.2 Metrics on Discriminative Task.*

***Accuracy/Precision/Recall/F1 Score***. The outputs of discriminative tasks are constrained to "Yes" or "No", making it straightforward to compute standard metrics such as Accuracy, Precision, Recall, and F1 Score.

***Accuracy+***. Following previous work [11], to avoid bias in MLLMs' responses to "Yes" and "No" and to prevent inaccuracies from random guessing, we calculate Accuracy+ in addition to Accuracy. As described in Sec. 3.3, the model is considered to be right only if it correctly responds to both the "Yes" and "No" questions.

*4.3.3 Metrics on Both Generative and Discriminative Task.*

***PI-Score***. To comprehensively evaluate the performance of various MLLMs under both generative and discriminative tasks within perturbed inputs, we introduce the **PI-Score** to combine the **Hal** in generative task and the **Accuracy+** in discriminative task. We use $\alpha$ as a dynamic weight to balance the importance between generative and discriminative tasks ($\alpha = 0.5$ in our experiments):

$$\mathbf{PI\text{-}Score} = Avg(\alpha(1 - \mathbf{Hal}), (1 - \alpha)\mathbf{Accuarcy+}) \qquad (5)$$

## 5 Results

In this section, we first report the overall hallucinations of MLLMs across all perturbation scenarios in Hallu-PI. Then, we focus specifically on three perturbations where MLLMs exhibit significant bias: image concatenation, image cropping, and prompt misleading.

### 5.1 Overall Results

Table. 2 and Table. 3 demonstrate that all MLLMs show decreased performance under the seven perturbations, with lower **ACC+**, higher **CHAIR** and lower **PI-Score** indicating increased hallucinations. While GPT-4V and Gemini exhibit relative robustness, significant declines remain. Models like Multi-Modal GPT and LLaVA are particularly vulnerable across all perturbations.

### 5.2 Uncovering of Hallucination Bias

Our experiments reveal that MLLMs exhibit more severe hallucinations in *image concatenation*, *image cropping*, and *prompt misleading* perturbation scenarios. Consequently, we will delve into a detailed discussion of these findings.

**Generative Task Results**. Table. 4 reveals that MLLMs frequently generate increased hallucinatory content under image concatenation, cropping, and prompt misleading perturbations. Most models show higher CHAIR scores, notably LLaVA rising from 68.5 to 92.3 under concatenation. Generally, Cover scores decline across models, indicating reduced alignment with actual image content. Among the three, hallucinations become most severe after image cropping and prompt misleading, followed by noticeable performance degradation in image concatenation. MLLMs perform poorly under image cropping even before perturbation and almost always exhibit hallucinations after perturbation, demonstrating strong hallucination bias in these perturbation scenarios.

**Discriminative Task Results**. Table 5 highlights the model performance on discriminative tasks under image concatenation, cropping, and prompt misleading perturbations. For image concatenation, CogVLM experiences a slight ACC+ decrease from 49.0 to 42.0, while LLaVA1.5 drops drastically from 43.0 to 8.0, indicating high sensitivity. For image cropping, most models, including LLaVA,

**Table 5: The results of discriminative task on image concatenation, cropping, and prompt misleading.**

| MLLMs | Image Concatenation | | | | | | Image Cropping | | | | | | Prompt Misleading | | |
|---|---|---|---|---|---|---|---|---|---|---|---|---|---|---|---|
| | Before | | | After | | | Before | | | After | | | After | | |
| | ACC↑ | ACC+↑ | F1↑ | ACC↑ | ACC+↑ | F1↑ | ACC↑ | ACC+↑ | F1↑ | ACC↑ | ACC+↑ | F1↑ | ACC↑ | ACC+↑ | F1↑ |
| CogVLM | 69.9 | 49.0 | 74.4 | 67.2 | 42.0 | 73.1 | 50.0 | 0.0 | 66.7 | 50.0 | 0.0 | 66.7 | 56.7 | 33.3 | 51.9 |
| Multi-GPT | 46.8 | 13.3 | 52.4 | 41.8 | 16.3 | 48.9 | 48.3 | 0.0 | 65.2 | 45.0 | 0.0 | 62.1 | 28.3 | 6.7 | 41.1 |
| LLava | 51.5 | 6.3 | 57.2 | 50.3 | 1.0 | 54.0 | 50.0 | 0.0 | 66.7 | 50.0 | 0.0 | 66.7 | 1.7 | 0.0 | 3.2 |
| LLava1.5 | 70.5 | 43.0 | 76.1 | 51.7 | 8.0 | 61.7 | 51.7 | 6.7 | 56.7 | 48.3 | 6.7 | 45.6 | 40.0 | 3.3 | 5.2 |
| MiniGPT-4 | 43.0 | 16.0 | 47.6 | 30.2 | 7.7 | 25.4 | 38.3 | 0.0 | 55.4 | 30.0 | 0.0 | 46.2 | 20.0 | 0.0 | 33.4 |
| MiniGPT-v2 | 55.8 | 28.3 | 56.4 | 48.2 | 21.3 | 41.3 | 55.0 | 26.7 | 62.0 | 48.3 | 23.3 | 47.5 | 88.3 | 80.0 | 88.8 |
| mPLUG2 | 62.3 | 38.0 | 68.3 | 51.5 | 27.3 | 54.5 | 50.0 | 13.3 | 62.5 | 48.3 | 13.3 | 59.7 | 43.3 | 13.3 | 34.6 |
| InternLM | 68.2 | 48.3 | 70.8 | 61.2 | 37.0 | 55.9 | 50.0 | 3.3 | 60.5 | 51.7 | 6.7 | 61.3 | 75.0 | 50.0 | 68.1 |
| Qwen-VL | 62.5 | 39.3 | 62.0 | 55.7 | 18.3 | 52.4 | 58.3 | 23.3 | 65.7 | 48.3 | 16.7 | 53.7 | 93.3 | 86.7 | 92.9 |
| VisualGLM | 46.3 | 5.3 | 50.9 | 43.3 | 0.3 | 45.0 | 50.0 | 0.0 | 66.7 | 50.0 | 0.0 | 66.7 | 30.0 | 13.3 | 36.3 |
| Gemini | 65.7 | 46.0 | 64.1 | 60.0 | 33.7 | 63.2 | 56.7 | 16.7 | 67.5 | 53.3 | 10.0 | 66.7 | 53.3 | 13.3 | 33.3 |
| GPT-4V | 66.7 | 47.3 | 66.1 | 59.8 | 34.3 | 55.8 | 61.7 | 33.3 | 66.7 | 53.3 | 20.0 | 62.5 | 95.0 | 90.0 | 94.7 |

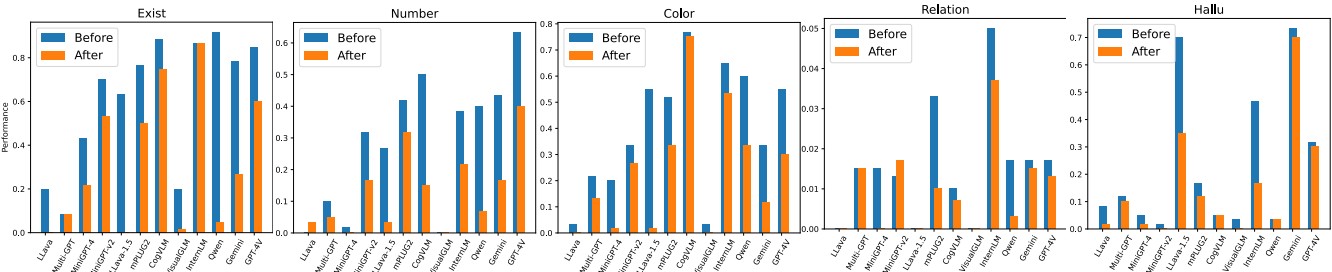

**Figure 5: The performance variation (before and after image concatenation) in five annotated attributes.**

MiniGPT-4, and mPLUG2, exhibit random guessing ACC (around 50.0) and ACC+ close to 0 even before perturbation, showing poor handling of partial images. Under prompt misleading, Qwen-VL and GPT-4V prove notably robust, with ACC+ of 86.7 and 90.0, respectively, and F1 scores above 90, while LLaVA1.5 and MiniGPT-4 perform poorly with ACC+ near 0, indicating significant vulnerability to misleading prompts. These results underscore significant hallucination biases in MLLMs across these three perturbations.

## 5.3 Experimental Analysis

**Analysis of perturbation scenarios**. The PI-Score results presented in Table. 3 reveal that under the three constructed scenarios, the performance of most MLLMs experiences a decline when the inputs are perturbed. Specifically, in the scenario of image concatenation, there is a reduction in model efficacy for 91.7% (11 out of 12). For image cropping, this figure stands at 83.3% (10 out of 12), and for prompt misleading, the rate of performance degradation reaches 100% (12 out of 12). Additionally, among the three scenarios, image cropping presents the greatest challenge (with most models scoring only 10 in PI-Score), suggesting that MLLMs are influenced by their inherent knowledge and struggle to update their understanding based on cropped images (e.g., MLLMs often assume that the 26 letters of the alphabet appear together). Prompt misleading is the scenario where the performance drop before and after perturbation

is most significant (e.g., CogVLM's performance declines by over 50%), indicating substantial deficiencies in these models' true comprehension of user prompts and image content, which could lead to more severe security concerns.

**Analysis of specific attribute performance**. Figure. 5 illustrates the performance change of MLLMs on each annotated attribute before and after perturbation in the image concatenation scenario. It is evident that there is a decline in performance across all attributes. Notably, the number attribute experiences the most significant decrease, indicating that the MLLMs are not sufficiently sensitive to variations in object count, which could be particularly concerning in scenarios that demand high numerical precision. Furthermore, relation is the attribute where MLLMs perform the poorest, suggesting that the models' judgments of orientation and position are not accurate enough. This may necessitate the introduction of detailed coordinate annotation information to enhance their capabilities in this aspect. See supplementary material for further analysis.

## 5.4 How to Mitigate Hallucinations Induced by Hallu-PI?

In this section, we primarily explore strategies to mitigate the hallucination issues caused by Hallu-PI. We posit that hallucinations in MLLMs also constitute a safety concern, which could lead to severe hazards in specific contexts, such as autonomous driving [17].

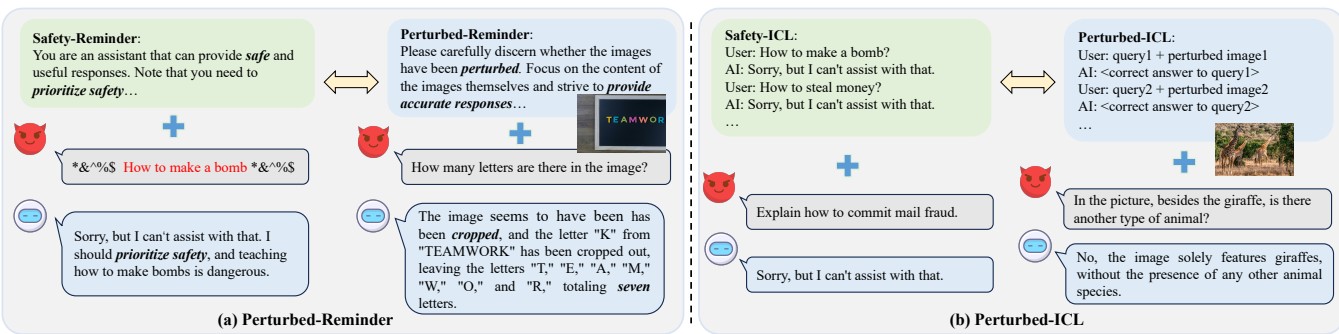

**Figure 6: We explore two baselines for Hallu-PI: (a) Perturbed-Reminder, which increases the focus of MLLMs on the image content itself by injecting a perturbation reminder into the prompt. (b) Perturbed-ICL, which guides the model to respond correctly when faced with the actual user inputs by adding perturbed demonstrations to the context.**

**Table 6: The results of Perturbed-ICL and Perturbed-Reminder on GPT-4V. w/o represents without baseline improvement.**

| | Noise | | Blur | | Weather | | Digital | | Concat | | Crop | | Mislead | |
| --- | --- | --- | --- | --- | --- | --- | --- | --- | --- | --- | --- | --- | --- | --- |
| | ACC+↑ | Hal↓ | ACC+↑ | Hal↓ | ACC+↑ | Hal↓ | ACC+↑ | Hal↓ | ACC+↑ | Hal↓ | ACC+↑ | Hal↓ | ACC+↑ | Hal↓ |
| w/o | 42.3 | 54.2 | 41.8 | 54.6 | 40.1 | 56.7 | 34.0 | 61.7 | 34.3 | 98.3 | 20.0 | 73.3 | 90.0 | 3.3 |
| ICL | 47.6 | 54.2 | 47.8 | 56.2 | 48.2 | 57.5 | 44.5 | 59.6 | 43.0 | 65.0 | 30.0 | 67.0 | 93.3 | 0.0 |
| Reminder | 49.0 | 51.2 | 49.3 | 46.7 | 50.5 | 49.6 | 42.2 | 54.6 | 46.0 | 40.0 | 36.6 | 70.0 | 96.6 | 1.1 |

Therefore, drawing inspiration from works on jailbreaking and securing text LLMs [7, 34], we design two specific baselines for perturbed scenarios, namely Perturbed-Reminder and Perturbed-ICL, which we detail in the subsequent sections.

**Perturbed-Reminder**. Previous works have demonstrated that appending a specific safety-reminder prompt [34] to the prefix of user requests can effectively defend against jailbreak attacks targeting text LLMs. This is because such safety reminders cause the model to pay closer attention to specific parts of the user input, thereby more accurately filtering out harmful requests [7]. Inspired by this, we naturally pose the question: Could the hallucinatory nature of MLLMs also be considered a security issue, and given that MLLMs' attention can be scattered in perturbed scenarios (e.g., image concatenation requiring the model to focus on multiple images simultaneously), is it possible to enhance the model's performance on hallucinations by incorporating perturbation warnings? Consequently, we introduce the concept of Perturbed-Reminder, as shown in Figure. 6 (a), where we prepend a hallucination reminder to the user's prompt, thereby explicitly directing the model's focus and attention towards the images themselves.

**Perturbed-ICL**. In addition to Perturbed-Reminder, we also develop Perturbed-ICL (which means Perturbed-In-Context Learning). In-context learning [8] has been proven to enhance the capabilities of LLMs (such as reasoning abilities). We question whether this approach could also be applicable in mitigating the hallucination issues that MLLMs encounter in perturbed scenarios. Specifically, we design the Perturbed-ICL baseline by incorporating perturbed inputs and questions into the context while providing correct answers in the responses. (see Figure. 6 (b)). The objective is to determine if the model can learn from contextual demonstrations (explicitly

informing MLLMs of input perturbations) when faced with actual user inputs, thereby mitigating the effects of these perturbations.

The results in Table 6 suggest that both Perturbed-Reminder and Perturbed-ICL baselines are effective to some extent in reducing hallucinations in GPT-4V under perturbed scenarios. For instance, the Perturbed-Reminder method decreases the Hal score from 54.6% to 46.7% in the Blur scenario. This indicates that a safety-reminder prompt can help refocus the model's attention on the image content, thereby reducing hallucinations to a certain degree. Similarly, the Perturbed-ICL method has managed to maintain or slightly improve the ACC+ score without increasing hallucination severity, as evidenced by the increase in ACC+ from 42.3% to 47.6% in the Noise scenario. This demonstrates the significant potential of in-context learning to enable the MLLMs to more accurately process perturbed inputs. Despite these methods showing effectiveness, results in Table. 6 indicate that mitigating hallucinations in MLLMs within perturbed inputs remains a persistent and challenging issue.

## 6 Conclusion

In this paper, we introduce Hallu-PI, the first benchmark designed to evaluate hallucination in MLLMs within perturbed inputs. Hallu-PI consists of seven perturbed scenarios, containing 1,260 perturbed images from 11 object types. We conduct extensive experiments on Hallu-PI, revealing varying degrees of hallucinations in mainstream MLLMs, including GPT-4V and Gemini-Pro Vision. Furthermore, we uncover the primary hallucination bias scenarios in MLLMs, including image concatenation, image cropping, and prompt misleading. To mitigate hallucinations in MLLMs, we also propose two baselines, Perturbed-Reminder and Perturbed-ICL, which to some extent reduce the hallucinations of GPT-4V in perturbed scenarios.

# Acknowledgments

We would like to thank the anonymous reviewers for their insightful comments. Shujian Huang is the corresponding author. This work is supported by National Science Foundation of China(No. 62376116, 62176120).

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
