# OpenReview forum: "Hallu-PI: Evaluating Hallucination in Multi-modal Large Language Models within Perturbed Inputs"
_acmmm.org/ACMMM/2024/Conference — MM2024 Poster_

### Official Review · Reviewer_Nbe3 · 2024-05-23

**Rating:** 5
**Confidence:** 4

**Summary:**

This paper proposes an experimental evaluation platform for assessing hallucinations in multilayer linear models under perturbed inputs. By constructing different perturbation scenarios to simulate various types of hallucinations, it is used to evaluate hallucinations in multi-modal large language models (MLLMs) across both generative and discriminative tasks.

**Strengths:**

1. This paper introduces a novel benchmark dataset for evaluating the hallucination generation capabilities of large multimodal language models under perturbed input conditions, thereby filling an existing gap in current benchmark suites which do not account for such distorted inputs.
2. This paper provides a comprehensive experimental evaluation and analysis of 12 widely-used large multimodal language models on the Hallu-PI benchmark, revealing the limitations and biases exhibited by these models when processing perturbed input data.
3. The authors propose two benchmark methods targeted at perturbed input data - Perturbed-Reminder and Perturbed-ICL.

**Limitations:**

1. Formatting Issues
  a. The spacing between Table 3 and the subsequent text appears excessive.
  b. The length of the expression X_obj in Section 4.2 exceeds the page margin boundaries.
  c. The second paragraph in Section 4.2 lacks a terminal punctuation mark at the end.
  d. The text "Prompt Misleading" in the second paragraph of Section 5.1 exceeds the page margin boundaries.
  e. The first row in Table 5 is not centered within the column.
2. Experimental Issues
  a. Table 5 does not provide the Before experimental results for the "Prompt Misleading" condition, preventing a comprehensive understanding of the impact of Prompt Misleading.
  b. This paper lacks sufficient explanation for instances where certain evaluation metrics increased after the perturbation was applied, such as the LLaVA1.5 results under the Weather condition in Table 2.

**Suitability:**

3

---

### Official Review · Reviewer_TAgv · 2024-05-24

**Rating:** 5
**Confidence:** 4

**Summary:**

The paper introduces Hallu-PI, a novel benchmark designed to evaluate hallucination in multimodal large language models. The key innovation is the implementation of a Perturbation operation on input images, serving as an effective measure of the model's robustness with respect to image integrity. Comprehensive assessments of state-of-the-art methodologies, such as GPT4V, Gemini-Pro, LLaVA-1.5, mPLUG-Owl2, and CogVLM, etc., have validated the effectiveness of the evaluation dataset.

**Strengths:**

1. The creation of a dataset that introduces perturbed inputs specifically for assessing hallucination in Multimodal Large Language Models (MLLMs) is an innovative approach to evaluation.
2. The proposed evaluation method and datasets are novel compared to existing hallucination evaluation datasets (e.g., PoPE, M-HalDetect, AMBER) with the different image perturbation strategies.
3. The paper is well written and easy to follow.
4. The evaluation of the current mainstream MLLMs are extensive, and the evaluation are convincing and consistent across various model.
5. It propose two potential methods to improve model's robustness in terms of perturbed inputs.

**Limitations:**

1. Will the evaluation code be available for further evaluation of MLLMs, which would benefit the research community?

2. Would it be beneficial to include additional perturbation operations such as 'cut-mix'[1], 'mix-up'[2] and 'shearing' for common image augmentation[3]? This could aid in understanding the robustness of each model tested as the Hallu-PI aims to test the robustness of MLLMs in terms of hallucination.

[1] Yun, Sangdoo, et al. "Cutmix: Regularization strategy to train strong classifiers with localizable features." Proceedings of the IEEE/CVF international conference on computer vision. 2019.

[2] Zhang, Hongyi, et al. "mixup: Beyond empirical risk minimization." arXiv preprint arXiv:1710.09412 (2017).

[3] Cubuk, Ekin D., et al. "Randaugment: Practical automated data augmentation with a reduced search space." Proceedings of the IEEE/CVF conference on computer vision and pattern recognition workshops. 2020.

3. It is not clear whether the metrics evaluated on Hallu-PI in Table 2,3,4 are stable or not, since the standard deviation of metrics are not reported. Because the MLLM's generation results would be varied when using different generation configuration (e.g., top-p, top-k sampling).

**Suitability:**

3

---

### Official Review · Reviewer_YiVb · 2024-05-25

**Rating:** 4
**Confidence:** 4

**Summary:**

This paper introduces a novel benchmark, Hallu-PI, designed to assess the hallucination performance of Multi-modal Large Language Models (MLLMs) under perturbed input conditions. Unlike previous benchmarks, Hallu-PI includes five types of perturbed scenarios, such as image cropping and noise, which reflect real-world disturbances. It contains 3,930 perturbed images from 11 object types, with detailed annotations and a rich set of questions suitable for both generative and discriminative tasks. The study reveals that mainstream MLLMs, including GPT-4V and Gemini-Pro Vision, exhibit significant hallucinations under perturbed conditions, a limitation not captured by standard unperturbed benchmarks. The paper also proposes two baselines, Perturbed-Reminder and Perturbed-ICL, to mitigate hallucination in perturbed scenarios. The results suggest that existing models need further improvements to handle perturbed inputs effectively.

**Strengths:**

1. The introduction of Hallu-PI is a significant advancement, as it is the first benchmark specifically designed to evaluate hallucination in MLLMs under perturbed inputs. This addresses a critical gap in the current evaluation methodologies.
2. The paper conducts extensive experiments on 12 mainstream MLLMs, revealing significant insights into their performance under perturbed conditions. This thorough evaluation provides valuable information on the current state of MLLMs and their limitations.
3. The proposal of two baseline methods, Perturbed-Reminder and Perturbed-ICL, to mitigate hallucinations in perturbed scenarios is a forward-thinking approach. These methods offer practical solutions to improve MLLM performance.

**Limitations:**

1. While the benchmark includes a variety of perturbations, there could be other real-world perturbations (e.g., extreme lighting conditions, occlusions) that are not covered. Expanding the types of perturbations could provide a more comprehensive evaluation.
2. The proposed evaluation metrics, such as PI-Score, provide useful insights, but they might not capture all aspects of model performance and hallucination severity. Can the authors evaluate the proposed method on Hallusionbench (CVPR'24) to evaluate the hallucination?

**Suitability:**

3

---

### Official Review · Reviewer_NWRM · 2024-05-27

**Rating:** 4
**Confidence:** 4

**Summary:**

The manuscript introduces a novel benchmark designed to assess the susceptibility of Multi-modal Large Language Models (MLLMs) to hallucinations when faced with perturbed inputs. The authors have crafted a comprehensive benchmark, Hallu-PI, which includes five distinct perturbation scenarios with a total of 3,930 images across 11 object categories. The paper presents an extensive experimental evaluation of 12 mainstream MLLMs, revealing significant hallucination biases, particularly in scenarios involving image concatenation, cropping, and prompt misleading. To address these issues, the authors propose two baseline strategies, Perturbed-Reminder and Perturbed-ICL, aiming to mitigate hallucinations in MLLMs. The study's findings underscore the need for further research to improve the robustness and reliability of MLLMs in real-world applications.

**Strengths:**

Innovative Benchmark: The creation of Hallu-PI fills a critical gap in evaluating MLLMs' performance under perturbed conditions, offering a valuable tool for future research.

Comprehensive Evaluation: The paper provides an exhaustive experimental analysis of various MLLMs, which is crucial for understanding their limitations and potential failure points.

Proposed Solutions: Introducing baseline strategies to mitigate hallucinations demonstrates a proactive approach to addressing the identified issues.

**Limitations:**

Generalizability Concerns: The benchmark's focus on specific perturbation scenarios may limit the generalizability of the findings. It would be beneficial to understand how the models perform across a broader range of perturbations.

Lack of Reliability Assessment: A critical weakness of the study is the absence of a reliability assessment for the proposed Hallu-PI benchmark. Without such an evaluation, it is challenging to determine the consistency and repeatability of the benchmark results, which are essential aspects of any scientific measurement tool.

**Suitability:**

3

---

### Meta-Review · Area_Chair_AL9x · 2024-06-25

**Recommendation:** Accept (Poster)
**Confidence:** 4

**Metareview:**

The paper received all positive ratings initially. The reviewers think the paper is well-motivated and the evaluation is comprehensive. The reviewers suggested that the authors add more empirical studies. The final decision is acceptance.